# Language Agnostic Speech Embeddings for Emotion Classification

**Apoorv Nandan** [1]   **Jithendra Vepa** [1]

## Abstract

In this paper, we propose a technique for learning speech representations or embeddings in a self supervised manner, and show their performance on emotion classification task. We also investigate the usefulness of these embeddings for languages different from the pretraining corpus. We employ a convolutional encoder model and contrastive loss function on augmented Log Mel spectrograms to learn meaningful representations from an unlabelled speech corpus.

Emotion classification experiments are carried out on SAVEE corpus, German EmoDB, and CaFE corpus. We find that: (1) These pretrained embeddings perform better than MFCCs, openSMILE features and PASE+ encodings for emotion classification task. (2) These embeddings improve accuracies in emotion classification task on languages different from that used in pretraining thus confirming language agnostic behaviour.

## 1. Introduction

Using self supervised learning to extract meaningful representations from large corpora has become popular in computer vision and language modelling, leading to remarkable improvements in the performance of deep learning models, with limited annotated data across a variety of tasks. A large unlabelled corpus of a particular language can be used to train an encoder like BERT (Devlin et al., 2019), which then produces condensed representations from input text. These representations, or embeddings, are easily fine tuned even with small amounts of labelled data for downstream tasks like text classification. Usage of similar procedures on speech data has also been proposed (Pascual et al., 2019; Ravanelli et al., 2020; Wang et al., 2020; Oord et al., 2018; Chung & Glass, 2020). The impact of such embeddings

---

[1]Oberve.AI, Bangalore, India. Correspondence to: Apoorv Nandan <apoorv@observe.ai>, Jithendra Vepa <jithendra@observe.ai>.

Published at the workshop on *Self-supervision in Audio and Speech* at the $37^{th}$ *International Conference on Machine Learning*, Vienna, Austria. Copyright 2020 by the author(s).

across tasks like Automatic Speech Recognition, speaker identification and emotion classification has been explored but their usefulness across different languages has not been analyzed.

Extracting good representations from speech also remains a challenging task, despite recent progress. Speech data is inherently complex, as it encodes characteristics of the speaker, prosody, language as well as the actual content being spoken. Modelling the sequential nature of audio also becomes arduous as the length of the sequence for even small segments of speech is of the order of a few thousand frames, compared to an average of 40 words per utterance of textual data. Individual frames in a speech segment are also less dense in terms of information than words in a sentence

We propose a new setup for learning representations from speech data using self supervised learning and explore the language agnostic behaviour of the learnt representations. The pretext task involves stochastic augmentation of Log Mel spectrograms from input speech and training an encoder network to extract relevant embeddings from them using contrastive loss. Our experiments show that self supervised learning of speech embeddings, in our setup, leads to representations that work well in languages different from the language used during pretraining. Fine-tuning the pretrained representations on a different language with small amounts of data leads to even better results on downstream tasks.

The major contributions of this work are as follows:
(a) We show a new framework for learning speech representations using self supervised training.
(b) We show that the learnt representations are language agnostic, as representations learnt from a speech corpus of one language boosts results on downstream tasks in another language.

## 2. Related Work

Several paradigms have come up for applying self supervised representation learning on speech data.

(Pascual et al., 2019; Ravanelli et al., 2020) use a convolutional encoder to learn features from raw waveforms and train it by using these features simultaneously for a variety of self supervised tasks like reconstruction of waveform in an autoencoder fashion, predicting the Log power spectrum,

Mel-frequency cepstral coefficients (MFCC) and prosodic features from raw waveforms. The encoder trains to extract representations that work jointly for all those tasks. These representations are then shown to improve performance on speaker identification, emotion classification and Automatic Speech Recognition tasks compared to popular speech features like MFCC and filter banks. Two versions of this encoder were named Problem Agnostic Speech Encoder (PASE) and PASE+ respectively.

(Chung & Glass, 2020) use autoregressive predictive coding (APC) to learn speech representations and show that these representations are useful for speech recognition, speech translation and speaker identification tasks.

(Oord et al., 2018) used contrastive predicitive coding (CPC) to learn representations that are useful for phone and speaker recognition tasks. Both these approaches try to predict information about a future audio frame $x_{k+n}$ using frame level information from a set of past frames $x_1, x_2, ..., x_k$. CPC aims to learn representations containing information that are most discriminative between $x_{k+n}$ and a set of randomly sampled frames. APC directly attempts to predict the future time frame via regression.

(Wang et al., 2020) masked segments of Log Mel spectrograms of speech and used a transformer model to reconstruct the masked portion to train their encoder. They show promising results on ASR tasks with Libri Speech and Wall Street Journal datasets.

CPC was also used by (Rivière et al., 2020) to create features that are effective at phoneme classification. They also showed that pretraining in one language can extract features that are useful even in other languages.

Most of these methods have been evaluated with a primary focus on speech recognition related tasks in a single language. We explore the use of our learnt representations across different languages so that they can benefit languages without large labelled corpora.

## 3. Method

The inherent challenges of modelling speech are further amplified in a supervised setting where large scale annotation is needed to improve performance. Self supervised learning solves this issue by defining a task where deterministic labels can be generated from the dataset without any external labelling. In our case, we do this by creating positive and negative pairs of embeddings, where positive pairs are formed by augmentation and processing of a single segment of speech and negative pairs are created from different speech segments. We train the network with the objective of maximising agreement between positive pairs of input and minimising the same between negative pairs. This objective

is materialised with a contrastive loss function explained in subsequent sections. We believe that solving this task will require learning relevant features of the input speech. These features should ideally be useful in other tasks as well.

### 3.1. Pretraining Setup

Our setup is inspired by *SimCLR* framework proposed by (Chen et al., 2020). The framework involves an *encoder network* $f(.)$, a projection head $g(.)$ and a module to augment Log Mel spectrograms extracted from speech input. (Figure 1)

The augmentation module randomly masks a block of frequency channels and a block of timesteps. This procedure has shown remarkable improvements in ASR WERs as shown by (Park et al., 2019). We use this process to produce two feature inputs from a segment of speech. These feature inputs are encoded into their representations using the *encoder network*. The projection head $g(.)$ transforms these representations into lower dimension output vectors where we apply contrastive loss. The purpose of this setup is to make the *encoder network* output representations that show different aspects of the input speech segment, like intonation, stress, rhythm and speaker characteristics. The augmented speech segments are encoded such that representations from stochastic augmentation of the same speech segment are in agreement with each other, and are different from the representations of other (different) speech segments.

#### 3.1.1. STOCHASTIC AUGMENTATION

The process for augmentation is described below:

1. Frequency Masking: We first choose the width $f_w$ from a uniform distribution of $0$ to a parameter $F_{max}$. If $N_f$ is the number of Mel frequency channels, we select a $f_0$ from $[0, N_f - f_w)$ and mask the channels $[f_0, f_0 + f_w)$.

2. Time Masking: Time masking is done in a similar fashion as frequency masking, by first choosing $t_w$ from $[0, T_{max}]$ and then choosing $t_0$ from $[0, N_t - t_w)$ and masking the timsteps $[t_0, t_0 + t_w)$. We always ensure that the maximum possible width $T_{max}$ is $p$ times the total timesteps $N_t$, where $p$ is tuned so that excessive loss of information does not happen.

The above augmentations have already been used in a supervised setup for improving ASR performance by (Park et al., 2019) and in an unsupervised setup for learning representations by reconstructing masked portions by (Wang et al., 2020).

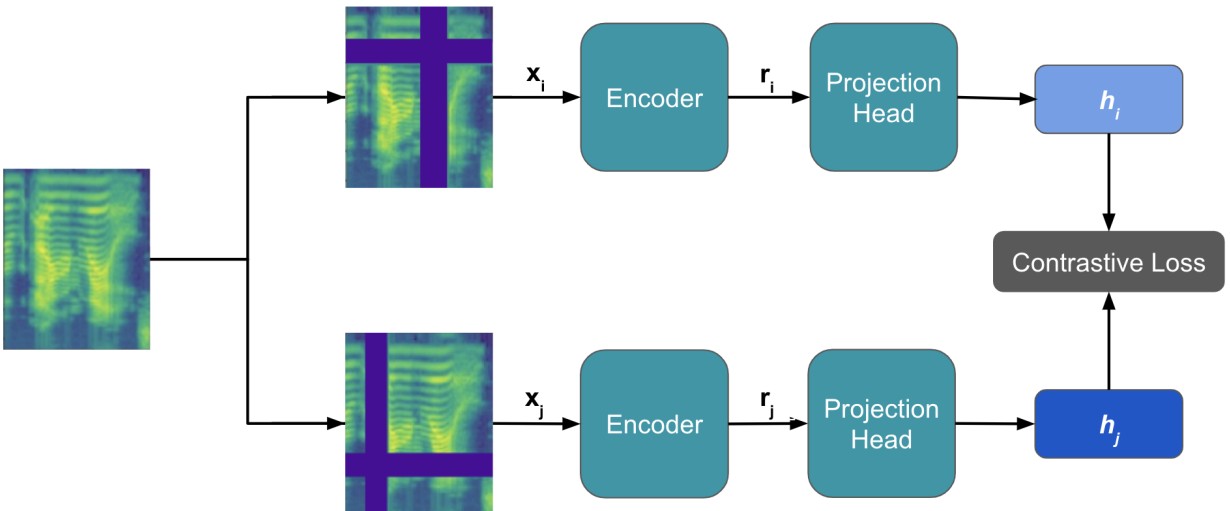

*Figure 1.* Schematic diagram of Contrastive Pretraining

### 3.1.2. ENCODER NETWORK

The *encoder network* is a convolutional neural network with architecture similar to the commonly used ResNet50 (He et al., 2016) network; 5 blocks of 3 convolutional layers with residual connections in each block. The output of this network after training serves as our embeddings, which are used as inputs for downstream task: Emotion Classification. We will refer to these embeddings as Contrastive Spec (CS).

$$r_i = f(x_i) \qquad (1)$$

### 3.1.3. PROJECTION HEAD

We apply two fully connected layers on the output of the *encoder network* to reduce it to a vector of dimension $d_h$.

$$h_i = W^{(2)}\sigma(W^{(1)}r_i) \qquad (2)$$

where $\sigma$ is ReLU activation function. These two layers, however, are discarded after pretraining and only $r_i$ in (1) is used as our embedding.

### 3.1.4. CONTRASTIVE LOSS

We use the *NT-Xent* loss function proposed by (Chen et al., 2020). Each input Log Mel spectrogram $x_k$ is augmented into two different matrices, and passed through the encoder network and the projection head to produce two different vectors $h_{2k-1}$ and $h_{2k}$. Thus, every pair $(2k-1, 2k)$ is a positive pair for our loss function (Chen et al., 2017). The rest of the pairs are negative. The loss for a given positive pair of inputs $(i, j)$ is:

$$l(i,j) = -\log \frac{\exp(\frac{h_i^T h_j}{\|h_i\|\|h_j\|\tau})}{\sum_{k=1}^{2N} \mathbb{1}_{[k \neq i]} \exp(\frac{h_i^T h_k}{\|h_i\|\|h_k\|\tau})} \qquad (3)$$

The total loss for a batch, of size $N$, is given by

$$\mathbb{L} = \frac{1}{2N} \sum_{k=1}^{N} [l(2k-1, 2k) + l(2k, 2k-1)] \qquad (4)$$

This loss is minimized by adapting the encoder weights to increase the cosine similarity between embeddings from a positive pair. For faster computation we cache the value of $s(i,j) = \exp(\frac{h_i^T h_j}{\|h_i\|\|h_j\|})$ for all pairs $(i, j)$ in the batch, and compute $\mathbb{L}$ above afterwards. For a batch size of $N$, we get $2N$ negative pairs per positive pair. $s(i, j)$ represents the similarity between $h_i$ and $h_j$. This loss function has also been used in (Sohn, 2016; Wu et al., 2018; Oord et al., 2018).

## 4. Downstream Task and Data

We use the following datasets for pretraining and evaluation of learnt embeddings.

- LibriSpeech (Panayotov et al., 2015) for training embeddings on our pretraining task. LibriSpeech is a popular corpus of English Speech consisting of over 1000 hours of speech recorded from audio books. We used the *train-clean-100* portion from it for our pretraining.

*Table 1.* Results on SAVEE dataset; accuracy for classification over 7 classes.

| FEATURES | CLASSIFIER | ACCURACY |
|---|---|---|
| MFCC | MLP | 41.7 |
| OPENSMILE | MLP | 43.6 |
| PASE+ | MLP | 50.8 |
| CONTRASTIVE SPEC. | MLP | **51.7** |

- Surrey Audio-Visual Expressed Emotion (SAVEE) database (Vlasenko et al., 2007). It consists of recordings from 4 male actors in 7 different emotions. The language is British English.

- German EmoDB (Burkhardt et al., 2005) for emotion classification. It contains about 500 utterances spoken in an emotional way by German actors.

- The Canadian French Emotional (CaFE) dataset (Gournay et al., 2018). It contains six different sentences, pronounced by six male and six female actors, in six basic emotions and two different intensities.

Features from openSMILE toolkit (Eyben et al., 2010) and Mel Frequency Cepstral Coefficient (MFCC) are two popular inputs for speech data. We use them to compute baselines performances.

Using a large dataset for English language during pretraining and finetuning it with a small amount of data from other languages emulates the scenario for a lot of languages where the available data is severely limited.

We evaluate the effect of pretraining embeddings using an English only dataset (LibriSpeech) on emotion classification task on English, German and French speech. We then fine tune our pretrained *encoder network* on a small amount of German speech and reevaluate the learnt embeddings.

## 5. Results

We first compare our learnt representations with standard speech features, and features from PASE+ encoder on SAVEE dataset (British English) for the downstream task of speech emotion classification. PASE+ encoder was trained on LibriSpeech dataset as well. The results are shown in Table 1.

The comparisons are done with a two layered feed forward network as the classifier. The hidden layer in the feed forward networks have 128 units with ReLU activations. The output layers use Softmax activations and the networks are trained with Cross Entropy loss. 13-dimensional MFCC features and 384-dimensional OpenSMILE features are used as baseline inputs.

Both pretraining strategies perform significantly better than MFCC features. PASE+ encoder converts raw samples into a higher-level representation, whereas we extract our embeddings from Log Mel spectrograms. We believe that modelling Log Mel spectrograms is simpler, and in hindsight, this may be part of the reason for better performance, along with the different pretraining strategy.

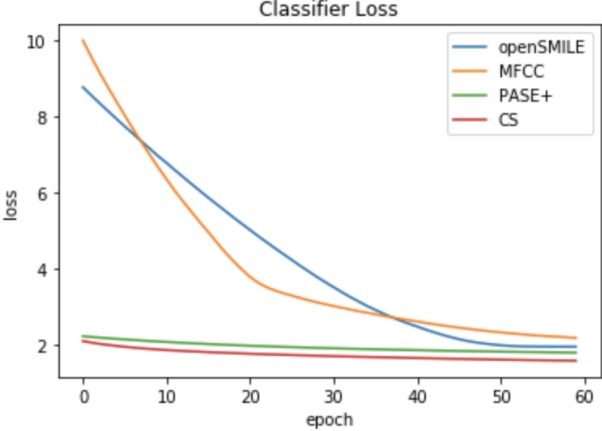

*Figure 2.* Validation losses for MLP classifier using openSMILE, MFCC, PASE+, and our embeddings as inputs for SAVEE dataset.

The benefits of embeddings over baselines features are also evident from the speed of convergence as shown by plots of validation loss of the classifier in Figure 2.

We evaluate the performance of these representations across different languages using French and German datasets for the task of speech emotion classification. We fit two simple models, Logistic Regression and a two layered feed forward network on the datasets using 13-dimensional MFCC features, 384-dimensional OpenSMILE features and embeddings from our pretrained encoder model (Zhang et al., 2016; Bachman et al., 2019; Kolesnikov et al., 2019). The details of this feed forward network are same as the classifier network used in previous comparison. The results have been summarised in Table 2.

Highlights from these evaluations are as follows:

- Our results show that pretraining in English outperforms regular speech features on the emotion classification task in not just English, but German and French datasets as well. (Table 2: *CS {Libri} Pretraining*)

- Finetuning the encoder model on small amounts of German speech further improves the performance. (Table 2: *CS {Libri, German} Pretraining*) This method of fine tuning the encoder on a small portion of data from the downstream dataset has been used in Natural

Language Processing as well to retrieve better representations from language models.

- A small feed forward network when trained with our embeddings as inputs, converges faster compared to baseline speech features. Figure 2 shows that pretrained embeddings achieve a low validation loss within few epochs. The embeddings make it easier to for the classifier to learn, which allows it to get a better performance quickly.

*Table 2.* Results on transfer across languages. In pretraining column, CS refers to Contrastive Spec. In Features, EMB refers to Embedding

| DATASET | PRETRAINING | FEATURES | LR ACC | MLP ACC |
|---|---|---|---|---|
| GERMAN EMO-DB | NONE | MFCC | 60.7 | 61.6 |
| | NONE | OPENSMILE | 63.7 | 64.2 |
| | CS {LIBRI} | EMB | 65.8 | 65.4 |
| | CS {LIBRI, GERMAN} | EMB | 66.6 | **67.4** |
| FRENCH EMOTIONAL DATASET | NONE | MFCC | 46.1 | 50.7 |
| | NONE | OPENSMILE | 51.1 | 51.3 |
| | CS {LIBRI} | EMB | 52 | **53.8** |
| | CS {LIBRI, GERMAN} | EMB | 54.2 | **55.1** |

## 6. Conclusion

We explored the effectiveness of self supervised training to learn representations from speech that are useful for emotion classification. We propose a setup where a convolutional encoder model to extract embeddings from Log Mel spectrograms using contrastive loss. After the pretraining is complete, even simple models are able to perform better than MFCC features and OpenSMILE features on emotion classification task. For emotion classification in English speech, we used SAVEE corpus. Our embeddings also perform better than PASE+ embeddings on this dataset.

Further, we found that representations learnt from one language are useful in other languages as well. We used embeddings from our encoder trained on English speech corpus, LibriSpeech, and observed that they outperform MFCC features in emotion classification with German and French speech. These evaluations were done on German EmoDB and CaFE corpus. We also observed that fine tuning our encoder model on German and French data, improves the performance.

Future work would include analysing the impact of augmentations on this procedure and the robustness of the learned embeddings. The transferability of these embeddings across different acoustic conditions can also be explored.

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
