# OpenReview forum: "Language Agnostic Speech Embeddings for Emotion Classification"
_ICML.cc/2020/Workshop/SAS — SAS 2020_

### Official Review · AnonReviewer2 · 2020-06-25
**Overall good paper, but probably written in a hurry. Several suggestions for improvement**

**Rating:** 7
**Confidence:** 3

**Review:**


The paper describes the application of self-supervised learning of speech embeddings to the task emotion classification. While the basic approach is not completely new, to best of my knowledge its application to emotion recognition, and language-independent emotion recognition is. Thus I believe the paper should be accepted.

Dear authors, it would have helped me be a little bit if you had been more precise in your description of the approach.
For example,
- Equation 3 confused me a bit as I couldn't find a definition of some of the symbols. I believe the ||z|| is a typo and should be ||h|| and probably the tau is a temperature parameter, and most likely the "j" in the denominator is another typo and should be a "k"?
- Figure 1 could be better to understand if you actually added some text what is what - where is "x", "f", where are the two fully connected layers W, where is h
- Equation 4, pls define N (batch size) when using it first time, not later in the text.
- Line 142, column 2: "." missing in "pair For"
- Line 143, column 2: again, the z in the equation seems to be a typo
- Line 157, column 2: "consistsing" typo
- Librispeech - there are different variants of it with different sizes. Please state what the size of the corpus you have used and which sets exactly (clean? other?)
- It would be good to know which features (or dimension of the feature vector) you have used from OpenSMILE
- Which MFCC implementation (or your own) you have used, number of coefficients, filterbanks etc. should be stated. Same for the Log Mel which you are using.
- Please mention the term "PASE+ encoder" in your literature review - a reader who is not familiar with it may be confused as there is no connection otherwise to the cited literature
- As you compare your approach to PASE+, you should give more details on how PASE+ differs from your approach - you only mention that you are using Log Mel and that your are better because of that. Doesn't the loss function also play a role?
- I miss a description of your MLP classifiers (topology, how have they been trained). As the extraction of the embedding already uses a CNN, the results of the experimental comparison of "MFCC+MLP" vs. "embedding+MLP" may depend on the number of parameters you spend in the MLP - maybe the MFCC+MLP  / OpenSMILE+MLP could catch up if you increase the size / number of layers of the MLP? A few experiments or better explanations could provide some more insight.
- Line 134/135 "which is" -> "which are"
- Line 147: "this layer", actually you are discarding 2 layers

---

### Official Review · AnonReviewer3 · 2020-06-28
**Contrastive loss based speech embeddings for language-independent downstream tasks**

**Rating:** 7
**Confidence:** 3

**Review:**

Summary
A new type of general / task-agnostic speech embedding is proposed that uses a data augmentation strategy and contrastive loss.  These embeddings are shown to contain language-independent information about emotion since an emotion classifier can learn from these embeddings, including for unseen languages, even without additional fine-tuning on the target language (although such fine-tuning gives additional improvement as well).  Comparison is made to MFCCs, openSMILE, and PASE+ features for an emotion classification task in English. Comparison is made to MFCCs and OpenSMILE for emotion classification in French and German.

Pros
* It is well-motivated to investigate cross-language ability of self-supervised speech embeddings.  The good results on a cross-language task show promise for applications in low-resource languages.
* good review of the relevant related works.
* published datasets are used, which is good for reproducibility.
* analyzing cross-language performance of speech embeddings is novel.

Cons
* While comparison to PASE is made in the same-language emotion classification task, PASE is conspicuously missing from the cross-language evaluation, even though PASE seems like the most interesting / relevant one to compare to.  The reader wonders how PASE performs in a cross-language scenario, and whether the gains of the proposed embeddings over PASE seen in the same-language task also generalize to the cross-language one.
* It would be nice to see cross-language performance on other downstream tasks besides emotion classification (although this is understood to be outside of the scope of this paper).

---

### Official Review · AnonReviewer1 · 2020-06-28
**Interesting work (and results) but with insufficient experimental validation**

**Rating:** 6
**Confidence:** 5

**Review:**

The authors document a study combining SpecAugment (Interspeech '19) with contrastive loss. The related work is thorough and very recent-- so recent and arxiv-heavy, that they should be revised, so that their listing refers to the actual venue, or mark as pre-print. Unfortunately, a pre-print from more than a year ago indicates, that the paper did not pass the peer review process, and should not be used as citation.

- Park et al., 2019 (SpechAugment) is Interspeech 2019.
- Oord et al,. 2018 (Representation learning): although I have very high regards for the authors, this paper seems to be not accepted for publication.
- Chen et al., 2020 (SimCLR) might still be under review?
- Devlin et al., 2018 (bi-directional Transformers for language understanding) seems to be not accepted for publication
- Pascual et al.. 2019 (problem-agnostic representations) is Interspeech 2019

The Method part is nicely written, just a tiny mistake in 3.1.4, Formula (3): In the denominator, j should be k.
They do a good job in summarizing their method for representation learning, however, they completely lack a description of their downstream classification method:

> We fit two simple models, Logistic Regression and a two layered feed forward network

is just not nearly enough detail on how the actual classification system is setup (layers? activations? loss? training schedule? ...). This is unfortunate, and maybe a mistake? There's enough space available on the last page, maybe the section was just forgotten?

Experimental section: Brief but informative. Tab 1 misses the logistic regression results. Results are promising.

...however, and this is my main concern here: Emotion classification is a rather niche task, and particularly on the "easy" version of acted speech. It would be more conclusive to show experiments on non-acted emotional speech (eg. Aibo), an ensemble of paralinguistic tasks (see Interspeech Challenges) or a better studied task such as ASR (in particular, since SpecAugment was LAS/e2e as opposed to a "classic" ASR system). The authors already trained on librispeech, so they could easily add that as a Baseline to show the effectiveness of the features. For multilingual ASR, they could have used europal or alike.

To sum it up: I think the paper matches the audience of the venue, is well written and shows good results. My main concerns are the missing description of the classification system (which could easily be added), the references (which can be fixed), and (and this is my personal opinion) baseline ASR experiments to support the effectiveness of the learnt representations -- emotion classification on acted data is not only niche, but also the "easiest" case.

---

### Decision · Program_Chairs · 2020-07-01

**Decision:**

Accept

**Comment:**

Dear author(s),

Thank you very much for your submission at the ICML2020@SaS workshop (https://icml-sas.gitlab.io/). Based on the scores assigned by the reviewers, we are happy to notify you that your paper was accepted for the workshop.

Please, address the comments of the reviewers and submit the camera-ready version by July 8. We ask the authors to record a 15min video for your talk. At the workshop, we will have the pre-recorded video as well as a live QA session. It is important to keep this time limit, otherwise, your talk will be automatically cut. The deadline for uploading the video is July 8. The detailed instructions for uploading will follow.

Feel free to contact us for any questions!

Best,

The ICML20@SaS organizers:
Mirco Ravanelli
Titouan Parcollet
Dmitriy Serdyuk
Devon Hjelm
Bhuvana Ramabhadran